# Acoustic Prompt Tuning: Empowering Large Language Models with Audition Capabilities

## Abstract

The auditory system plays a substantial role in shaping the overall human perceptual experience. While prevailing large language models (LLMs) and visual language models (VLMs) have shown their promise in solving a wide variety of vision and language understanding tasks, only a few of them can be generalised to the audio domain without compromising their domain-specific capacity. In this work, we introduce **A**coustic **P**rompt **T**urning (APT), a new adapter extending LLMs and VLMs to the audio domain by soft prompting only. Specifically, APT applies an instruction-aware audio aligner to generate soft prompts, conditioned on both input text and sounds, as language model inputs. To mitigate the data scarcity in the audio domain, a multi-task learning strategy is proposed by formulating diverse audio tasks in a sequence-to-sequence manner. Moreover, we improve the framework of audio language model by using interleaved audio-text embeddings as the input sequence. This improved framework imposes zero constraints on the input format and thus is capable of tackling more understanding tasks, such as few-shot audio classification and audio reasoning. To further evaluate the reasoning ability of audio networks, we propose natural language audio reasoning (NLAR), a new task that analyses across two audio clips by comparison and summarization. Experiments show that APT-enhanced LLMs (namely APT-LLMs) achieve competitive results compared to the expert models (i.e., the networks trained on the targeted datasets) across various tasks. We finally demonstrate the APT's ability in extending frozen VLMs to the audio domain without finetuning, achieving promising results in the audio-visual question and answering task. Our code and model weights will be released.

## 1 Introduction

Auditory stimuli contribute to shaping the overall human perception experience. While visual language models (VLMs) (Li et al., 2023; Liu et al., 2023; Dai et al., 2023; Shukor et al., 2023; Han et al., 2023) that are capable of solving diverse down-streaming tasks have emerged driven by the advent of large language models (LLMs) and massive visual-text pretraining, only few of them (Shukor et al., 2023; Han et al., 2023) can be adapted to the audio domain while maintaining their performance in the image/video domain.

To embrace more than two modalities, few works recently attempted to explore the diversity and heterogeneity of tasks and modalities. UniVAL (Shukor et al., 2023) unified input/output format, model architecture, and training objective, and therefore, learned a shared encoder-decoder LLM with multi-modal curriculum learning. ImageBind-LLM (Han et al., 2023) adopted ImageBind, a cross-modal encoder bundling six modalities (including images) to a shared embedding space, and adapted the LLM with a frozen image encoder. While both works extended visual LLMs to other domains, in addition to the considerable amount of training data, they are bundled to a specific architecture, hindering the ability to adapt them to a new modality.

Meanwhile, following the VLM framework, a few works proposed audio-only LLMs where a pair of audio clip and text token are used as inputs for text generation. LTU (Gong et al., 2023) bridged audio with language modalities by end-to-end finetuning on an instruction-based dataset. Pengi (Deshmukh et al., 2023) applied multi-task learning to leverage off-the-shelf datasets, alleviating the data-scarcity issue. Still, they are restricted to two domains (i.e., audio and language). They also cannot address

tasks beyond the [audio, question, answer] format, e.g., few-shot audio classification (Liang et al., 2023). One question thereby arises: *Can we adapt LLMs/VLMs to the audio domain by simply encoding sound clips as acoustic prompts?*

In this work, we introduce APT (**A**coustic **P**rompt **T**uning), an acoustic adapter that extends LLMs and VLMs to audio understanding and reasoning tasks using soft prompts only. Specifically, APT encodes audio clips into audio feature maps and then uses an audio aligner to generate acoustic prompts conditioned on both input instructions and the audio feature maps. When training APTs, a multi-task learning strategy is adapted by formulating diverse audio tasks in a sequence-to-sequence format. Besides popular audio tasks (such as audio tagging and audio captioning), APT makes full use of publicly-available datasets by training on three new tasks, namely query-based sound event detection, temporal event retrieval, and sound event counting, to learn fine-grained audio features. In addition, we improve the audio language model framework by juxtaposing acoustic prompts with text embeddings. Rather than applying soft prompts as a prefix to the input texts, the improved framework exerts no constraints on the format of the input sequence. Therefore, the APT-enhanced LLMs, namely APT-LLM, can analyse multiple audio clips in a single feed-forward process, facilitating more audio understanding tasks, such as few-shot audio classification and audio reasoning. To further evaluate models' reasoning ability, we propose a new task referred to as natural language audio reasoning (NLAR) which is devised to distinguish, compare, and summarise two audio clips. Experiments on existing audio understanding tasks, including audio tagging, audio captioning, and few-shot audio classification, show that APT-LLM achieves performance on par with those obtained by audio language models or even domain-expert models. APT also yields a good performance on the proposed NLAR, indicating its capacity to comprehend over a single audio clip. Finally, quantitative studies are conducted to demonstrate that APT improves the performance of a VLM in the audio-visual question and answering (AVQA) task.

Our contributions are summarized as below:

- An acoustic adapter is introduced to extend LLMs and VLMs to the audio modality by soft prompting. To mitigate data scarcity in the audio domain, we improve the present multi-task training approach by devising new tasks and their corresponding prompts during training. Leveraging the annotations in off-the-shelf databases, APT-LLM learns acoustic embeddings with fine-grained features from task discrepancy.

- APT formulates diverse audio tasks as a sequence-to-sequence task where generated text is conditioned on interleaved audio-text tokens. Without any constraints on the input format, APT-LLM is not only able to solve different tasks according to the diverse instructions, but also to exploit the correlation among different audio clips in the same sequence. To the best of our knowledge, APT-LLM is the first audio-language model reasoning beyond a single audio clip.

- Natural language audio reasoning, a new audio comprehension task, is proposed to distinguish, compare, and summarise two audio clips. Compared to existing audio tasks, this new task not only evaluates model ability to understand an audio clip, but also requires models to analyse the content of two recordings by comparison and summarisation. APT-LLM is then benchmarked on this task.

- BLIP-2 (Li et al., 2023) coupled with APT (namely APT-BLIP-2) is studied qualitatively and quantitatively on the audio-visual question and answering task (Yang et al., 2022). Without further finetuning, APT-BLIP-2 can work with the visual modality directly, showcasing an efficient approach for extending multi-modal LLMs to a new modality.

## 2    RELATED WORKS

**Multimodal language models.** From recent advances, LLMs (Touvron et al., 2023; Chiang et al., 2023; OpenAI, 2023) has exhibited astonishing comprehending and reasoning capacity. Driven by the open-world knowledge in LLMs, a variety of visual language models have been proposed with different alignment methods to integrate image/video data to text tokens (Alayrac et al., 2022; Li et al., 2023; Dai et al., 2023; Zhang et al., 2023b). However, most of them are restricted to the visual domain, largely due to the lack of training data in other domains (such as audio) and modality discrepancies. Recently, ImageBind-LLMs (Han et al., 2023) bridged the image encoder of

ImageBind (Girdhar et al., 2023), a six-modality language model, with an LLM and used visual tokens as soft prompts within the language model. UniVAL (Shukor et al., 2023) uniformed the input/output, the architecture, and the training object of multimodal LLMs and then devised a curriculum learning for gradual exposure to new modality. While both works adapted VLMs to other domains, they demands massive multimodal data to train the overall networks from scratch. Instead, this work investigates a domain-specific adapter that can be applied to extend any existing VLM/LLM to an additional modality (such as audio).

**Audio language models.** Following VLM, some works built audio language models for sound-only tasks. SpeechGPT (Zhang et al., 2023a) collected a speech-text instruction dataset, thereby learned to perceive and generating speech content in the audio. LTU (Gong et al., 2023) rendered an open-end dataset, containing 3.7M [audio, question, answer] tuples, and learned with a perception-to-understanding curriculum. While the aforementioned models achieved a good audio comprehension ability, they required a uniform input format as a triplet tuple. To work around this question, Pengi (Deshmukh et al., 2023) proposed a multi-task framework where an audio language model is trained with off-the-shelf audio datasets by prompted with different predefined questions. This work differs from these prior works in three-fold: 1) Rather than an audio-only language model, APT explores how to adapt existing VLMs and LLMs to the sound domain; 2) APT-LLM improves the multi-task framework by designing three new training tasks. By accessing existing datasets from different aspects, APT-LLM learns a fine-grained audio representation, and 3) APT-LLM re-frames the present input format, namely [audio, question, answer], to let audio and text alternate in a sequence. In this way, APT-LLM is able to ingest more than one audio clip in a single feed-forward, unleashing it to more audio tasks. To the best of the knowledge, APT-LLM is the first model that integrates in-context learning with multi-task training.

## 3 METHOD

Current audio LLMs (Gong et al., 2023; Deshmukh et al., 2023) learned to bridge audio with language by framing popular audio tasks (e.g., classification and captioning tasks) to the audio-conditioned text generation problem. Going beyond the [audio, question, answer] format, APT-LLM encodes multiple audio clips in one feed-forward process and juxtaposes them with text embeddings without any order constraint. The more flexible training paradigm mitigates the need for high-quality data and massive databases, and thus reduces required computations. Moreover, juxtaposing audio clips with texts enables APT-LLM to address more comprehensive reasoning tasks, such as natural language audio reasoning. We first discuss the overall architecture of APT-LLM in Section 3.1, and then elaborates APT-LLM learning objective in Section 3.3 and the training recipe in Section **??**. In Section 3.4, we define the natural language audio reasoning task, a new task to evaluate the audio comprehension ability of models.

### 3.1 ARCHITECTURE

The overall structure of APT-LLM is illustrated in Figure 1, with main components of an audio encoder, an audio aligner, and a large language model. APT-LLM alternates audio clips with text tokens without any format constraints and thus benefits from task diversity and large-scale pretraining.

**Audio encoder: from spectrograms to feature maps.** We use Audio-MAE (Huang et al., 2022), a vanilla 12-layer transformer encoder that learns to reconstruct randomly-masked spectrogram patches during training, as the audio encoder. Rather than using the last layer that finetuned for classification tasks, we apply the output feature map from the penultimate block of an Audio-MAE to encode fine-grained patterns in the sound.

**Audio aligner: from 10-second feature maps to a fixed number of audio tokens.** This module connects the audio encoder to the frozen language model as shown in Figure 1. It ingests a text prompt together with a variable number of audio feature maps extracted by the audio encoder as input and produces a fixed number of acoustic embeddings. Following the implementation of (Li et al., 2023), four transformer blocks constitute our audio aligner where 32 trainable embeddings attend to the input text tokens and extract the relevant information from the audio feature maps. Resampling a varying-length of audio embeddings to 32 acoustic embeddings, APT aligner reduces

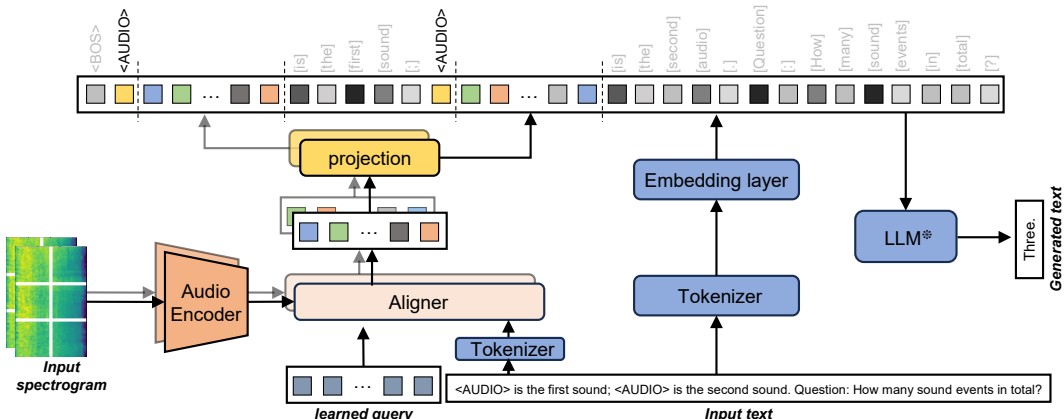

Figure 1: Illustration of the proposed APT-LLM. APT-LLM is constituted by three components: an audio encoder, an audio aligner, and a LLM. The audio encoder extracts audio feature maps from input spectrograms. The audio aligner then projects each audio feature map to 32 acoustic embeddings according to the input text. These acoustic embeddings, together with the added embeddings of the audio token "<AUDIO>", are juxtaposed with text embeddings. The interleaved audio-text embeddings are fed into the LLM to generate the output text. APT-LLM can ingest multiple audio clips in a sequence and thus benefit from diverse tasks during training.

the computational complexity in following attention mechanism while filtering out the information irrelevant to the input text.

**Large language model: from interleaved audio-text tokens to generated text.** The language model predicts the output text by taking into account the previous generated texts and the input audio-text tokens. We freeze all parameters in the language model during training. In addition to existing works (Li et al., 2023; Gong et al., 2023), we add before each audio clip a learnable audio token, "<AUDIO>", as a special token to indicate the beginning of audio tokens. We find this token helps the language model to distinguish audio tokens from text tokens when interleaving them together.

## 3.2 LEARNING OBJECTIVE

In order to motivate our training framework, we first present the learning objective used in existing work (Deshmukh et al., 2023). Let an audio-text pair in [audio, question, answer] format be referred to as $(a, t, g)$ where $a, t, g$ are the audio clip, input text, and output text, respectively, and $\mathbf{X}$ be input sequential embeddings to the language model. To align the audio modality to the language modality, an audio encoder $\mathcal{A}$ and an audio aligner $\mathcal{M}$ project the audio $a$ into a sequence $\mathbf{X}_{\text{audio}}$:

$$\mathbf{X}_{\text{audio}} = \mathcal{M}_\theta(\mathcal{A}_\phi(a, t)), \tag{1}$$

where $\phi$ and $\theta$ are the parameters of the audio encoder $A$ and the aligner $\mathcal{M}$. The audio embeddings are used as a prefix and then concatenated with the input text embeddings as

$$\mathbf{X}_{\text{audio;text}} = \mathcal{C}(\mathbf{X}_{\text{audio}}, \mathbf{X}_{\text{text}}) = \mathcal{C}(\mathcal{M}_\theta(\mathcal{A}_\phi(a)), \mathcal{W}_\psi(t)), \tag{2}$$

where $\mathcal{C}$ is a concatenating function and $\psi$ denotes the parameters of the word embedding layer $\mathcal{W}$ in the language model. Assuming the length of the concatenated embeddings $\mathbf{X}_{\text{audio;text}}$ be $L$, the parameters of the audio LLM are optimised by measuring the probability distribution of the next token conditioned on its previous tokens:

$$p(\mathbf{X}_{\text{pred}}|\mathbf{X}_{\text{audio}}; \mathbf{X}_{\text{text}}) = \prod_{i=L+1}^{L+|g|} p_{\phi,\theta,\psi}(\mathbf{x}_i|X_{\text{audio;text},<i}; \mathbf{X}_{\text{pred},<i}), \tag{3}$$

In this way, prevailing LLMs are able to unify many audio-to-text tasks in a sequence-to-sequence manner. However, not all understanding tasks can be fitted into the format of [audio, question, answer] (e.g., to learn a new concept using a handful of labelled audio examples), calling for a new paradigm that can exploit diverse tasks in a uniform input/output format.

Table 1: Multi-task learning strategy adopted by APT-LLMs. "#Audio samples" denote the number of audio clips in the dataset. Stage 0-2 denotes audio-text alignment, learning from single audio clips, and learning from multiple clips, separately.

| Task | Training stages | | | Dataset | #Audio samples | Durations | Setup |
|------|:---:|:---:|:---:|---------|:-------------:|:---------:|:-----:|
| | 0 | 1 | 2 | | | | |
| Audio tagging | ✓ | ✓ | ✓ | AudioSet | 2M | 5.8kh | train/test |
| Audio captioning | ✓ | ✓ | ✓ | Wavcaps AudioCaps Clotho v2 | 400k 39k 7k | 7.6kh 108h 31h | train/test |
| Audio question and answering | | ✓ | ✓ | Clotho AQA | 2k | 12h | train |
| Query-based sound event detection | | ✓ | ✓ | AudioSet-SL | 81k | 226h | train |
| Temporal event retrieval | | ✓ | ✓ | AudioSet-SL | 81k | 226h | train |
| Sound event counting | | ✓ | ✓ | AudioSet-SL | 81k | 226h | train |
| Few-shot audio classification | | | ✓ | AudioSet | 2M | 5.8kh | train/test |
| Natural language audio reasoning | | | ✓ | NLAR | 0.2k | 1.2h | train/test |

We thereby propose a new learning framework in which interleaved audio-text embeddings are used as the LLM's input such that the model is able to leverage and learn from more diverse tasks during training. Let $\mathbf{a}$ and $\mathbf{t}$ be audio clips and input text, and $g$ still be output text. Assuming both $\mathbf{a}$ and $\mathbf{t}$ have $N$ different elements, the input audio-text pairs are denoted as $[(a^i, t^i)]_{i=1}^N$ where $a^i$ and $t^i$ are the $i$-th audio clip and input text, respectively. Eqn. (2) can be re-written as

$$\mathbf{X}_{audio;text} = \mathcal{I}(\mathbf{X}_{audio}, \mathbf{X}_{text}) = [\mathcal{M}(\mathcal{A}_\phi(a_1, t_1)), T_\psi(t_1), \dots, \mathcal{M}(\mathcal{A}_\phi(a_N, t_N)), T_\psi(t_N)], \quad (4)$$

where $\mathcal{I}$ is the function that alternating acoustic embeddings with text embeddings. In this way, APT-LLM can integrate multiple audio clips in the input sequence, enabling itself to learn from more audio understanding tasks.

### 3.3 MULTI-TASK LEARNING STRATEGY

With the uniform input/output format, APT-LLM is able to learn from a large variety of audio tasks and thus benefiting from the diverse training datasets. As shown in Fig. 1, instead of passing through all training data directly, APT-LLM is trained through:

**Audio-text alignment.** Before coupled with a LLM, we pretrain APT audio aligner to bridge the audio modality and the text modality. To this end, we freeze the other components and optimise parameters of the audio aligner with audio-text pairs from AudioSet (Gemmeke et al., 2017) and WavCaps (Mei et al., 2023). During training, a fixed number of acoustic embeddings are learnt to extract relevant information from the audio feature maps according to the input text tokens. Following Li et al. (2023), the audio aligner learns with triplet training objectives: audio-text matching, Audio-grounded text generation, and audio-text contrastive (See more in Appendix A.1)

**Learning from single audio clip.** After APT has extracted acoustic embeddings according to the input text, the following LLM learns to project these tokens to the word embeddings of the targeted LLM. APT-LLM is thus trained with multiple tasks using various prompts (see more in Appendix A.2). In addition to existing audio tasks, namely audio tagging, audio captioning, and audio question and answering, we design three new tasks: (1) *Query-based sound event detection* that aims to train a model to predict the onset and offset time of a specific sound event; (2) *Temporal event retrieval* that is to recognise sound events occurred in a specific period, and (3) *Sound event counting* that requires a model to count the frequency of a specific sound event in a recording. Instead of rendering datasets, we exploit the publicly-available AudioSet with strong labels (Hershey et al., 2021) using different prompts (see more in A.2). This multi-task framework facilitates APT-LLM's learning from diverse datasets, including AudioSet (Gemmeke et al., 2017), WavCaps (Mei et al., 2023), AudioSet with strong labels (Hershey et al., 2021), Clotho (Drossos et al., 2020), AudioCaps (Kim et al., 2019), and Clotho-AQA (Lipping et al., 2022).

Table 2: An example demonstrating APT-LLM's capacity of audio reasoning. It requires audio networks to comprehend recordings and reasoning across multiple recordings.

| Natural Language Audio Reasoning (NLAR) example: *"Where is the sudden sound?"* |
|---|

User

Wav1:"AmbianceBackyard_Quiet_bip.wav"   Wav2:"Rain hitting window.wav"

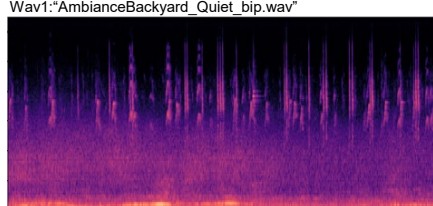 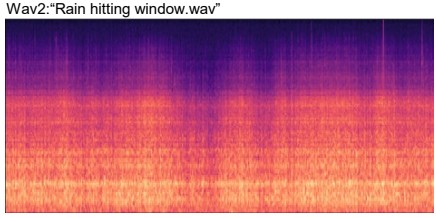

Question: Which recording has a more sudden and startling sound event?

| APT-LLM | First. |
|---|---|
| Ground truth | first |

**Learning from multiple audio clips.** In addition to the aforementioned tasks, APT-LLM learns from two additional tasks by juxtaposing more than one audio clips with input text. Specifically, few-shot audio classification and natural language audio reasoning are added to the multi-task training framework in this stage. On the one hand, for the few-shot audio classification, APT-LLM predicts labels of sound events by exploiting the correlation between input audio clips. On the other hand, APT-LLM is required to compare and summarise two different sounds in the natural language audio reasoning task (see the following Section 3.4). Trained on these two tasks, APT-LLM learns to analyse beyond a single recording and answer questions as per input questions. We adopts AudioSet (Gemmeke et al., 2017) and the proposed datasets for few-shot audio classification and natural language audio reasoning, respectively.

## 3.4    NATURAL LANGUAGE AUDIO REASONING TASK

One of the complex reasoning ability of human is to learns across different pieces of sounds, understanding what happening in each audio and analysing the content of different audio by comparison and summarisation. However, existing audio tasks focus on analysing acoustic scenarios in an independent recording by recognising the inside sound events Kong et al. (2020) and/or retrieval their spatio-temperal information Politis et al. (2022). We thus propose natural language audio reasoning (NLAR), a new task where the model is required to answer questions by explicitly comparing or summarising two different audio recordings. Table 2 showcases an example in the NLAR task. An audio system takes two audio clips together with a free-form text query as input and is expected to answer the question by taking into consideration the both audio. For details of the design process and examples of the proposed natural language audio reasoning task, please refer to Appendix A.3. Compared to existing audio tasks, the proposed audio reasoning task features three notable differences:

**Comprehension of multiple audio clip**: This task requires a model to answer open-ended questions by comparing or summarising the content of two different audio clips. The model must first comprehend the two audio clips as per the raised question separately and answer the question by taking into account the two audio inputs. An example of the audio reasoning task can be found in Table 10.

**Diverse question types**: Questions for natural language audio reasoning task assess diverse auditory aspects, such as the presence, the frequency, and acoustic features of sound events. Therefore, the model should not only ground the sound events in the recordings, but also retrieve relevant information as per the input question.

**Effects of the chronological order**: Compared to existing audio tasks, e.g., (Li et al., 2023) and (Gong et al., 2023), the proposed audio reasoning task emphasises the order of the audio recordings in a sequence. In other word, the answer associated with the audio pair "[Audio A, Audio B]" could be different with the answer associated with "[Audio B, Audio A]" when their questions are the same.

Table 3: Zero-shot performance comparison with audio language models. We group the methods in terms of their training strategy. "#Params." denotes the number of trainable parameters and "#Pairs" represents the number of audio-text pairs. ↑ indicates the higher number, the better performance.

| Model | #Params. | #Pairs | AudioSet (mAP↑) | AudioCaps (SPICE↑) | Clotho (SPICE↑) |
|---|---|---|---|---|---|
| *Audio-language models trained with the contrastive loss* | | | | | |
| AudioCLIP (Guzhov et al., 2022) | 30M | 2M | 25.9 | - | - |
| CLAP (Elizalde et al., 2023) | 190M | 128k | 5.8 | - | - |
| *One-for-all models for various audio tasks* | | | | | |
| LTU (Gong et al., 2023) | 96M | 5.7M | 18.5 | 17.0 | 11.9 |
| Pengi (Deshmukh et al., 2023) | >191M | 3.4M | - | 18.2 | 12.6 |
| APT-LLM | 101M | 2.6M | 14.7 | 17.1 | 11.6 |

In this way, we expect audio understanding models to be able to attend to different portions of the input sequence when the question vary.

By evaluating audio language models on the natural language audio reasoning task, we achieve more comprehensive assessment of audio language models.

## 4 EXPERIMENTS

APT was first coupled with LLMs (i.e., APT-LLM) and evaluated as a general-purposed audio learner on a variety of existing audio-related benchmarks, including audio tagging, audio captioning, and few-shot audio classification. To further assess its ability in comprehending two audio clips of interest, APT-LLM was further benchmarked on the natural language audio reasoning task. In addition to audio comprehension, we also experimented and analysed (quantitatively and qualitatively) APT as an zero-shot adapter to BLIP-2 (Li et al., 2023; Dai et al., 2023), a state-of-the-art VLM.

### 4.1 EXPERIMENT SETUP

Our models were implemented relying on the BLIP-2 framework (Li et al., 2023). We used Audio-MAE (Huang et al., 2022) as the audio encoder in all APT models we developed. Considering Audio-MAE only contains 100M parameters, we used a two-layer transformer as the aligner to bridge the audio and text domains. Without an explicit statement, we coupled APT with Vicuna 7B v1.1 (Chiang et al., 2023) for evaluation. We testified APT-LLM with two close-ended datasets: AudioSet (Gemmeke et al., 2017) and ESC-50 (Piczak, 2015); and four open-ended datasets: Clotho (Drossos et al., 2020), AudioCaps (Kim et al., 2019), natural language audio reasoning (in Section 3.4), and audio-visual question and answering (AVQA) (Yang et al., 2022).

Adam optimiser was used for model training. We applied Warmup strategy in the first 2K steps and used a cosine linear learning rate in the following steps. We trained the APT models using three NVIDIA A100 (40G) GPUs. The audio-text alignment pretraining and multi-task training took 5 days separately.

### 4.2 COMPARISON WITH EXISTING APPROACHES

We compare APT-LLM against the state-of-the-art specialised systems (i.e., the networks trained with task-specific data) and previous audio language models on existing tasks, including audio tagging, audio captioning, and few-shot audio classification.

**Audio tagging** requires models to predict classes of test samples from a predefined label set. We evaluated the models on the AudioSet dataset (Gemmeke et al., 2017). During inference, APT-LLM was prompted using the sentence "*Summarize the audio with key words.*" Since APT generates free-form texts directly, we used the APT text encoder pretrained in the stage 1 to encode generated answers and the given classes names to text embeddings. Afterwards, cosine similarity is calculated as the classification probably. Consistent with the findings in previous work (Gong et al., 2023), Table 4 shows a performance gap between audio language models and task-specific models. This is expected since the latter addresses the classification task as a close-end problem, with much lower complexity than open-ended problem where models need to search across the entire word embedding

Table 4: Performance comparison in audio captioning tasks. ↑ indicates the higher number, the better performance.

| Model | AudioCaps | | Clotho | |
|---|---|---|---|---|
| | SPICE ↑ | SPIDEr ↑ | SPICE ↑ | SPIDEr ↑ |
| *Specialised systems trained with task-specific examples* | | | | |
| PANNs-BART (Xu et al., 2021) | 0.153 | 0.183 | 0.083 | 0.127 |
| CNN-GPT2 (Kim et al., 2023) | 0.167 | **0.438** | 0.111 | 0.215 |
| WSAC+PD (Kouzelis & Katsouros, 2023) | 0.173 | 0.403 | 0.123 | 0.247 |
| *One-for-all models for various audio tasks* | | | | |
| APT-LLM | **0.191** | 0.402 | **0.132** | **0.248** |

space. In addition, we found that the performance of the text encoder greatly impacts the classification result when evaluating the generated answers. This finding can be explained by the fact that word embeddings of different classes should be sparse enough to when measuring their distance to the embeddings of generated answers.

**Audio captioning** is the task where models are supposed to generate free-form description according to an input recording. The sentence "*Describe the audio clip concisely.*" is applied as the input prompt. We finetune APT-LLM two epochs on the training split of AudioCaps (Kim et al., 2019) and Clotho (Drossos et al., 2020) datasets and compare it with the captioning models trained on the both tasks. As shown in Table 4, APT-LLM achieves the best performance on both AudioCaps and Clotho datasets in terms of SPICE and SPIDEr.

Table 5: Accuracy (%) of various methods on ESC-50 in the few-shot settings.

| | Accuracy↑ | |
|---|---|---|
| | 5-way | 12-way |
| *Specialised systems trained with task-specific examples* | | |
| ProtoNet (Snell et al., 2017) | 88.2 | 77.7 |
| MatchNet (Vinyals et al., 2016) | 86.8 | 71.8 |
| HPN (Liang et al., 2022) | 88.7 | 78.7 |
| *Audio language models trained with constractive learning* | | |
| TIP-adapter (Zhang et al., 2022) | 97.5 | 95.6 |
| Treff adapter (Liang et al., 2023) | 98.5 | 96.3 |
| *One-for-all models for various audio tasks* | | |
| APT-LLM | 91.0 | 54.2 |

**Few-shot audio classification** is to classify test audio clips using labelled audio examples. Models were evaluated in the $N$-way $K$-shot problem where: (1) there are $N$ classes in the classification task, and (2) each class contains $K$ different audio examples. Following previous works (Liang et al., 2023) in this task, we tested APT-LLM in the 5/12-way 5-shots settings. In our free-form query design, we prompt the few shot classification question by adding the query audio clip together with the input text "*This is a sound of*" to the end of the sequence of labelled audio examples and their corresponding label texts. We implemented the same evaluation protocol to all few-shot learners for a fair comparison. As shown in Table 5, APT-LLM outperforms the task-specific models (Snell et al., 2017; Vinyals et al., 2016; Liang et al., 2022) in the 5-way 5-shot setting while having a competitive performance compared to CLAP Adapters (Zhang et al., 2022; Liang et al., 2023). In the 12-way 5-shot problem, however, we can observe a performance degradation of APT. We suspect this may be due to the limitation of attention mechanism in LLMs when addressing very long sequences (12-way 5-shot modelling results in a sequence of roughly 2420 tokens). It should be noted that while APT-LLM was trained with 4-way 1-shot tasks, it can generalise to other few-shot settings, suggesting that APT-LLM learns to act as a few-shot classifier rather than memorising the expected answers.

### 4.3 EVALUATION ON NATURAL LANGUAGE AUDIO REASONING

Since APT-LLM is able to ingest multiple audio clips in a single feed-forward process, we investigated APT-LLM with natural language audio reasoning for which a model is expected to distinguish, compare, and summarise two audio clips (see Appendix A.3). To the best of the knowledge, there is no previous work evaluating model ability to comprehend more than one recording. We thus contrast APT-LLM to the baseline where predictions are fixed to a specific answer (we used "yes" as the fixed answer after sev-

Table 6: Benchmarking APT on the natural language audio reasoning task.

| Model | Accuracy↑ (%) |
|---|---|
| the baseline | 29.9 |
| APT-Vicuna v1.1 | 62.9 |
| APT-Vicuna v1.5 | **63.8** |

eral attempts). Table 6 demonstrates that APT-Vicuna v1.5 achieves 63.78% mAP score, outperforming the baseline by a large margin. This result suggests that APT-LLM is able to not only comprehend the content in an audio clip but also analyse more than one audio recordings by comparison and summarisation. It is worth noting that there is marginal improvement when upgrading Vicuna from v1.1 to v1.5, indicating the performance of language models is not the bottleneck in this task, at least for the two used in our study.

### 4.4 EVALUATION ON ZERO-SHOT AUDIO-VISUAL TASKS

APT was also experimented as an audio adapter for an existing VLM, BLIP-2 (Li et al., 2023). BLIP-2 consists of a frozen image encoder, a Qformer, a projection layer, and a frozen Vicuna v1.1. Therefore, we integrated the APT trained with the same language model to BLIP-2 by interleaving acoustic prompts with text embeddings. We refer to the APT-enhanced BLIP-2 as APT-BLIP-2. Of note, although we selected BLIP-2 as our backbone model, APT can

Table 7: Performance comparison between different modalities in audio-visual learning.

| Model | Modal | Accuracy↑ |
|---|---|---|
| BLIP-2 (Li et al., 2023) | Video-only | 42.9 |
| APT-LLM | Audio-only | 27.7 |
| APT-BLIP-2 | Audio-video | **59.7** |

be easily adapted to another language model. APT-BLIP-2, together with other multimodal language models, was investigated on a audio-visual question and answering dataset where models are expected to choose one out of four options by using both audio and video modalities. We experimented APT-BLIP-2 on the subset of the AVQA dataset (Yang et al., 2022) as many video links associated with the AVQA test segmentation were no longer available on the internet at the time of the experiment. As shown in Table 7, APT-BLIP-2 yielded a better performance than video-only and audio-only counterparts, indicating the adaptation to the audio domain benefits models' learning from the content of video.

### 4.5 LIMITATIONS

In this work, we devised APT to align acoustic embeddings with text embeddings of language models. Now that the word embeddings change when switching to a different language model, even if their architectures remain the same, each language model calls for a dedicated APT for adaptation. In addition, APT-LLM was not trained with instruction-based datasets, and thus, has limited ability to response to questions excluded from the training set. Finally, we purposely focused APT-LLM training and experimentation on general-purpose audio understanding tasks, therefore, unlikely it can understand speech and music audios.

## 5 CONCLUSIONS

We proposed APT, a general-purpose acoustic adapter that extends LLM/VLM to the audio domain. We showed that LLM coupled with APT is a multi-task audio learner that not only achieved a competitive performance across various audio understanding tasks but also be capable of in-context learning when fed with a few labelled examples. We also benchmarked APT-LLM's audio comprehension ability via the natural language audio reasoning task, a new task that requires a model to distinguish,

compare, and summarise two different audio clips. Last but not least, it is evident from our study on audio-visual learning that encoding sound clips as word tokens is an efficient approach to adapt LLM/VLM to the audio domain. Future works can extend audio language models into comprehension of music and speech audio, and make them more aligned with human perception via instruction tuning. It is also interesting to investigate how audio language models handle errors in the in-context learning.

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

# A APPENDIX

## A.1 AUDIO-TEXT ALIGNMENT

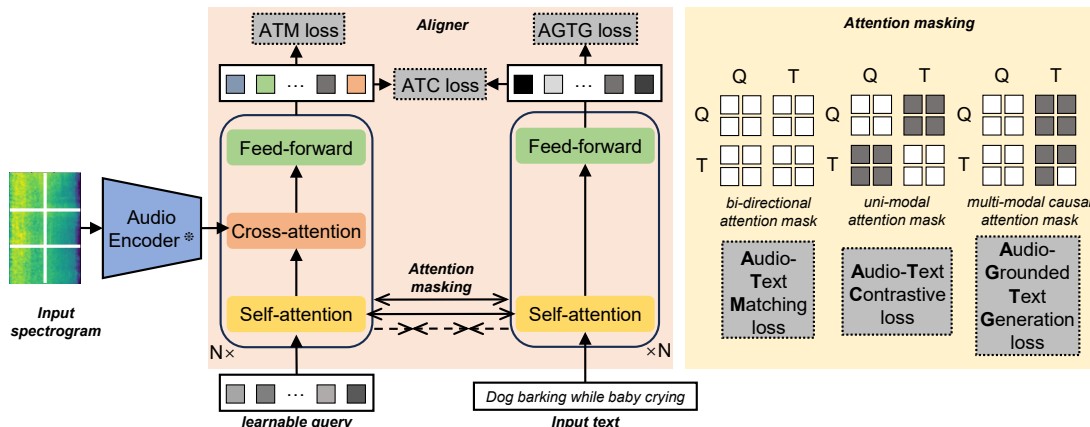

Figure 2: Illustration of APT audio aligner pretrained with audio-text pairs. The rest components are frozen in the pretraining. Using different self-attention masks, the parameter of the audio aligner is optimised with triplet learning objectives: Audio-Text Matching (ATM), Audio-Grounded Text Generation (AGTG), and Audio-Text Contrastive (ATC).

Before coupled with a LLM, we pretrain APT audio aligner to bridge the audio modality and the text modality. Figure 2 shows the implementation of audio-text alignment pretraining. Following Li et al. (2023), we freeze the other components and optimise parameters of the audio aligner. During training, a fixed number of acoustic embeddings are learnt to extract relevant information from the audio feature maps according to the input text tokens. In the pretraining, the audio aligner learns with triplet training objectives:

**Audio-text matching (ATM).** The goal of ATM is to learn a token-level alignment between acoustic and text tokens. APT is trained to distinguish whether a pair of an audio recording and a text sentence are from the same source. A bi-directional self-attention mask is applied to the input sequence such that acoustic and textual tokens can attend to each other. APT's output embeddings $Z$ thus carry both audio and text information. Each vector of its embeddings $Z$ is then fed into a binary classifier, and the prediction is produced by averaging the binary logits over all vectors.

**Audio-grounded text generation (AGTG).** AGTG shapes APT's learning to generate texts, given the input audio tokens. The information for generating the text is first extracted from the audio tokens, and then passed to the text tokens via self-attention layers. In APT, a multimodal causal self-attention mask is used to control audio-text interaction, where each text token can attend to all audio tokens and its previous text tokens.

**Audio-text contrastive (ATC).** The ATC enforces alignment between audio representation and text representation such that their mutual information is maximized. It achieves so by contrasting the audio-text similarity of a positive pair against those of negative pairs. APT aligns acoustic embeddings outputed by the audio encoder with text tokens. To avoid information leak, we employ a unimodal self-attention mask, where the acoustic and text tokens are not allowed to see each other.

## A.2 PROMPTS FOR VARIOUS TASKS

Tables 8 and 9 detail the prompts we used to prompt APT-LLMs during training and inference.

## A.3 NATURAL LANGUAGE AUDIO REASONING

Natural language audio reasoning (NLAR) is proposed to evaluate model's ability in comprehending across audio clips as per the text instruction. We progressively render the NLAR dataset using the Clotho-AQA dataset, and every stage is described in the following:

Table 8: Templates of the question and answer for audio tagging and audio captioning.

| audio tagging | audio captioning |
|---|---|
| Summarize the audio with key words. | Summarize the audio succinctly. |
| What sound events can be heard in the audio clip? | Present a short overview of the provided audio samples. |
| What auditory incidents can be recognized in the recording? | Provide a compact summary of the auditory content. |
| Which auditory occurrences can be detected? | Offer a brief outline of the audio clips that have been given. |
| Which sound occurrences can be perceived? | Render a compressed version of the audio's main points. |
| Present a concise breakdown of the given audio clips. | Describe the audio clip concisely. |
| List the sound events in the audio clip. | Explain the audio clip in a brief and straightforward manner. |
| Describe the recording with names of sound events. | Write a terse but informative summary of the sound. |
| Enumerate the audio events present in the audio. | Give a quick overview of the provided audio excerpts. |
| Name the auditory incidents in the audio sample. | Outline the given audio samples briefly. |
| #Output: {LABEL} | #Output: {CAPTION} |

- **Data collection**: We filter out some audio samples from the Clotho-AQA by the quality of the audio and the understandably of the annotations. To avoid data leakage, the test split of NLAR is collected from the test split of Clotho-AQA only;

- **Data cleaning**: The annotation in the Clotho-AQA is noisy. For instance, annotators might not reach an agreement on the number of bird chirping in the recordings. Therefore, we manually re-annotate the audio files again by focusing on the "controversal" annotations. We notice that it is fallible to calculate the frequency of sound events in some cases, and even we can not annotate their frequency, such as raining. In this case, we will annotate the presence of these activities throughout the recordings by using tags such as"in the beginning", "in the middle", "in the end" and "throughout the recording".

- **Describe acoustic features**: We applied ChatGPT-turbo (OpenAI, 2023) to describe the acoustic feature of each sound event with the prompt "describe the acoustic characteristic of {SOUND} precisely with a sentence less than 10 words" where {SOUND} refers to as the name of the sound event. We inspect each audio features to ensure they are comprehensible.

- **Create audio-text pairs**: We applied ChatGPT-turbo to generate five question-answer pairs according to their associated sound events together with the temporal information, and audio description of the two audio recordings.

- **Inspect the rendered data**: We manually check the generated question-answer pairs using the annotations of the recordings and call ChatGPT again if the number of qualified pairs is lower than five. We dropped the generated pairs only if they contains some factual errors. We noticed that during data rendering, some answers to the questions could be "unclear".

Table 9: Template of the question and answer for query-based sound event detection, temporal event retrieval, and sound event counting.

| query-based sound event detection | temporal event retrieval | sound event counting |
|---|---|---|
| Pinpoint the presence of {LABEL} with the time stamps. | Summarize the audio with key words in the interval of {STT} seconds to {EDT} seconds. | How many times can the sound {LABEL} be heard? |
| Indicate the start and end time of the audio event {LABEL}. | What sound events can be heard from {STT} seconds to {EDT} seconds? | How many instances of the sound {LABEL} can be perceived? |
| Document the exact times the sound {LABEL} taking place. | What auditory incidents can be recognized in the recording from {STT} seconds to {EDT} seconds | What is the number of times the sound {LABEL} is detectable? |
| Specify the time stamps for {LABEL} occurrence. | Which auditory occurrences can be detected during {STT} seconds to {EDT} seconds? | How frequently can one hear the sound {LABEL}? |
| When the sound {LABEL} happens? | Which sound occurrences can be perceived between {STT} seconds and {EDT} seconds? | How often can the sound {LABEL} be perceived? |
| Capture the exact times when {LABEL} is happening. | Present a concise breakdown of the recording from {STT} seconds to {EDT} seconds. | |
| Describe the time intervals during which {LABEL} takes place. | List the sound events in the interval of {STT} seconds to {EDT} seconds. | |
| State the precise moment at which {LABEL} occurs. | Name the auditory incidents within the {STT} to {EDT} seconds timeframe. | |
| What time does the sound event {LABEL} take place? | Enumerate the audio events present between {STT} seconds and {EDT} seconds. | |
| Capture the beginning and end time of the sound {LABEL}. | Describe the recording with names of sound events within the {STT} to {EDT} seconds timeframe. | |
| #Output: {STT}s-{EDT}s | #Output: {LABEL} | # Output: {NUMBER} |

We didn't exclude this from the dataset as it is also important for models to learn what cannot be perceived from the audio.

## A.4 ANALYSIS ON FEW-SHOT AUDIO CLASSIFICATION

Figure 3 (a) and (b) show APT-LLM performance in various few-shot settings. In both settings, the input sequence becomes longer when the number of shots/classes increases. It is interesting to observe that as the number of shots increases, the overall performance becomes worse. This is contrasting to the behavior of few-shot learners based on back propagation. One possible reason is that increasing the number of classes/shots rapidly increases the length of the input sequence, which is detrimental to APT-LLM global attention mechanism.

Table 10: An example of rendering the NLAR dataset

Based on the following two audio clips, generate 5 different questions that must be derived by summarising or comparing both audios and is prohibited to contain the information indicating its answer. The following information is provided: the sound events appear in the audio clip, together with its acoustic features, and corresponding onset and offset time stamps or frequency in the recordings. The answer should be either a binary one that can be responded by 'yes' or 'no', or an open-ended one that can be reply by a number or a single word.

Audios:
 First:
  wood creaking (rustic, rhythmic, and creaky) [6 times];
 second:
  thunder (explosive, rumbling, and reverberating) [1 times];
  rain (gentle, pitter-patter, rhythmic) [throughout the recording].
Questions and answers:
 1. are there 6 wood creaking and 2 thunder sounds in total? - no
 2. how many wood creaking or thunder are there in total? - 7
 3. does the second recording more likely to make one feel calm? - yes
 4. in which recording the events are more frequent? - first
 5. the frequency of wood creaking in the first recording is 6 times
  more than the frequency of thunder in the second one. - yes

Audios:
 First:
  tapping glass (crisp, clear, and tingling sound) [9 times];
 second:
  water (rapid, and draining water sound) [throughout the recording].
Questions and answers:
 1. are there 9 tapping glass sounds and 3 water sounds in total? - no
 2. how many tapping glass or water sounds are there in total? - 10
 3. does the second recording create a continuous sound throughout? - yes
 4. in which recording are the sound events more repetitive? - first
 5. does the second recording sound more dynamic compared to the
  first recording? - yes

Audios:
 First:
  wood creaking (rustic, rhythmic, and creaky) [6 times];
 second:
  shower (droplets, soothingly cascading) [throughout the recording].
Questions and answers:

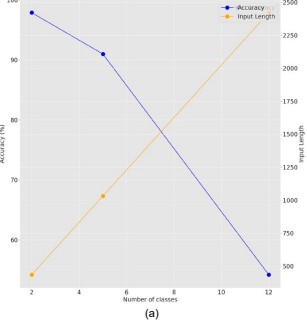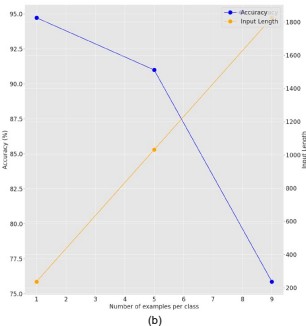

Figure 3: APT-LLM performance in few-shot audio classification when: (a) the number of classes increases, and (b) the number of examples per class increases. In both cases, the increase in the number of classes/examples leads to the rapid increase of the input sequence's length.

