# OpenReview forum: "Acoustic Prompt Tuning: Empowering Large Language Models with Audition Capabilities"
_ICLR.cc/2024/Conference — Submitted to ICLR 2024_

### Official Review · Reviewer_wWds · 2023-10-30

**Soundness:** 3 good
**Presentation:** 4 excellent
**Contribution:** 2 fair
**Rating:** 5
**Confidence:** 4

**Summary:**

Acoustic prompt tuning (APT) is proposed in this paper, which integrates an Audio-MAE encoder, a Q-Former aligner, and a Vicuna LLM to implement a multimodal LLM that can hear and understand audio events. Three auxiliary loss functions are used to improve the training of the Q-Former aligner, including audio-text matching (ATM), audio-grounded text generation (AGTG) and audio-text contrastive (ATC). APT-LLM is evaluated on audio tagging, audio captioning, few-shot audio classification, audio-language reasoning and zero-shot audio-visual question-answering tasks.

**Strengths:**

1. The paper investigates the integration of LLM with an audio encoder to empower it with auditory abilities, which is currently an understudied research problem.

2. The method is evaluated on a number of audio-event-related tasks evaluating the abilities of audio-event understanding and reasoning.

3. The presentation of the paper is very clear, including a precise limitation section that elaborates on the scope of the approach.

4. In-context learning for audio-event classification is investigated.

**Weaknesses:**

1. The authors claimed "Diverse audio-related tasks are formulated in a sequence-to-sequence manner without imposing any constraints on input sequences", which is not true since the input sequence could not be speech or music, as claimed in the limitation section. There might also be a maximum input sequence length imposed by the use of the Audio-MAE encoder, and the Q-Former aligner.

2. The authors claimed that one of the key contributions of the paper is: "this is the first attempt to unify fully-supervised learning with in-context learning." It is not clear to me what this means precisely. The authors need to make the motivation and benefits of combining multitask training with in-context learning clear.

3. The performance of the proposed approach is not satisfying based on Table 2, in particular on audio tagging on the AudioSet dataset.

4. The model is not tuned to follow instructions and can only perform a small of tasks, which makes the use of LLM less reasonable.

5. It sounds less sensible to me to use a standard Q-Former to convert audio input sequences into a fixed number of 32 tokens. Generating a fixed number of tokens is a good choice for images with fixed sizes, but not so much for audio input sequences due to their variable input lengths.

6. It is not clear what are the benefits of having the ATM, ATC, and AGTG multitask training on the Q-Former aligner.

**Questions:**

1. What's the size of the Vicuna LLMs used in the paper?

2. What are the strengths of APT-LLM compared to other recent methods, such as LTU? It seems APT-LLM has worse performances based on Table 1.

---

> ### Author Response · Authors · 2023-11-23
> **Response to Reviewer wWds**
>
> Thank you for your valuable feedback! We address your comments one by one as follows:
>
> [Maximum input sequence length]
>
> Although AudioMAE ingests fixed-length waveforms, the APT-LLM supports audio clips of variable length as input. Specifically, we will truncate or pad audio to 10 seconds length and generate acoustic tokens for each 10-second audio clip. After that, we will concatenate the acoustic tokens together so that the APT-LLM can digest varying-length audio clips.
>
> Regarding “ without imposing any constraints on input sequences”, we replace it with “impose zero constraints on input format” to avoid potential misunderstanding.
>
> [Benefits of combining multitask training with in-context learning]
>
> Thanks for your valuable feedback. We have updated the contribution part for clarity.
>
> In terms of the benefits of the combination, as shown in the Table 3, fewer audio-text pairs and parameters are introduced to the overall network compared to the existing approaches while APT still yielded a comparable performance compared to other audio-language models.
>
> [Performance on audio tagging and captioning]
>
> We notice that compared to the common classifiers, there is a performance drop due to the difficulty for the open-set issue and text matching. For example, when CLAP, a open-set classifier pretrained with contrastive loss, is used for evaluation, it calculates the similarity with 527 classes whereas APT predicts an appropriate token out of 32k tokens. Another issue is how to match the predicted answers with the ground-truth labels. Suppose APT outputs “rat” and we get the class of “mouse” in the ground truth. In that case, a proper text matching approach is required for a fair comparison. During evaluation on audio classification tasks, we used the APT’s checkpoint from audio-text-alignment pretraining as the text encoder. However, we did find that using different text encoders to evaluate the same checkpoint impacts the classification values a lot. We are planning to apply some discriminative text encoders, such as CLIP text encoder and ChatGPT Ada model, for the evaluation, and we will add the discussion on text matching to the Appendix.
>
> Regarding the audio captioning task, training on multiple datasets does not improve the performance on a specific dataset because the model cannot fit the dataset-specific vocabulary. In addition to the general-purpose audio [1], we found the same phenomena in other domains [2,3]. Following [3,4], we have conducted additional experiments to finetune APT on both Clotho and AudioCaps datasets for two epochs only. We compared the finetuned APT with expert models trained on the same datasets. As shown in Table 4, the APT yielded the best scores on both tasks.
>
> [Why LLM?]
>
> It is very tempting to build up foundational models by prompting LLMs: a) LLM can understand diverse texts as input such that we don’t have to design a special token for each task, e.g., [5]; b) LLMs have the ability to reasoning, facilitating us to explore more audio tasks, such as natural language audio reasoning; c) since some works [6-7] support that GPT/transformer can be a good in-context learner, LLMs can help the model figure out the correlations among input features in the few-shot settings.
> Given LLMs can bring the above advantages,  it would be tempting to explore how to adapt LLMs in the audio domain.
>
>
> [Architecture design]
>
> Regarding “Generating a fixed number of tokens is not so much for audio input sequences due to their variable input lengths”, as we mentioned in “[Maximum input sequence length]”, it is actually 10-second audio clip corresponds to a fixed number of tokens. We have updated the manuscript for a better understanding.
>
> Regarding the triplet losses in the audio-text alignment pretraining, we did a few preliminary studies on the pretraining and decided to use the combination for a better alignment.
>
> [Size of Vicuna]
>
> We used Vicuna 7B throughout the work. We have clarified this in the paper.
>
> [Strength of APT than other approaches]
>
> APT highlights three features compared to the other approaches:
> a) Training efficiency: as shown in Table 1, APT uses fewer training data and roughly 100M learnable parameters while yielding a competitive performance than other approaches;
> b) Audio understanding and reasoning across recordings: APT improves the framework of audio language model by juxtaposing acoustic and text embeddings as inputs. This facilitates APT digesting multiple audio clips, enabling it to tackle more audio tasks;
> c) As a audio adapter, APT extends LLM/VLM without optimizing the parameters of language models and losing their original abilities as an LLM/VLM. Our experiments show that APT can work with VLM without any audio-visual learning.

---

> > ### Comment · Reviewer_wWds · 2023-11-23
> > **Comments to the authors' response**
> >
> > I would like to thank the authors' response. However, the responses could not resolve my concern. It also comes very late (~1 hour before the deadline) and therefore prohibits further discussions. As a result, I prefer to keep my original rating but encourage the authors to revise the paper according to the suggestions from the reviewers.

---

> ### Author Response · Authors · 2023-11-23
> **Response to Reviewer wWds [Reference only]**
>
> [1] PREFIX TUNING FOR AUTOMATED AUDIO CAPTIONING
>
> [2] LLARK : A MULTIMODAL FOUNDATION MODEL FOR MUSIC
>
> [3] SIMVLM: SIMPLE VISUAL LANGUAGE MODEL PRETRAINING WITH WEAK SUPERVISION
>
> [4] BLIP-2: Bootstrapping Language-Image Pre-training with Frozen Image Encoders and Large Language Models
>
> [5] AudioPaLM: A Large Language Model That Can Speak and Listen
>
> [6] Why Can GPT Learn In-Context? Language Models Secretly Perform Gradient Descent as Meta-Optimizers
>
> [7] Transformers learn in-context by gradient descent
>
> P.s. Sorry, we have to make the reference a new comment due to the limited characters.

---

### Official Review · Reviewer_cBEu · 2023-11-01

**Soundness:** 2 fair
**Presentation:** 2 fair
**Contribution:** 2 fair
**Rating:** 3
**Confidence:** 3

**Summary:**

This paper describes Audio Prompt Tuning (APT), a new model architecture that interleaves representations of audio with embeddings of text tokens in order to enable joint learning of text-generation tasks that are conditioned on one or more audio clips and related text prompts. When coupled with a pretrained LLM, APT forms the APT-LLM model architecture, capable of performing a variety of audio-text tasks including audio tagging, audio captioning, and few-shot audio classification. In addition, the authors created a new type of tasks called natural-language audio reasoning (NLAR), in which the model is tasked with answering natural-language questions concerning the relations between two audio clips (but see my concerns below).

**Strengths:**

S1. Proposing a novel way of representing an arbitrary number of audio clips and their associated textual information that interleaves representations of the audio clips with text token embeddings. This significantly enhances the versatility of the model, giving it abilities to perform a variety of audio-text bimodal tasks such as tagging, captioning, classification, as well as tasks that involve more than one audio clip such as NLAR.
S2. A multi-task training recipe for the APT model that encompasses different cross- and self-attention mechanisms between the audio input and associated text input.

**Weaknesses:**

W1. The levels of accuracy achieved by APT-LLM on the audio tagging and audio captioning are unfortunately slightly disappointing (Table 1. Section 4.2). They fall below the accuracy levels of domain-expert models, by a noticeable margin in the case of audio tagging. The authors argue that this has to do with the open-ended nature of APT-LLM's output, which puts APT-LLM at a disadvantage compared to the closed-ended nature of the previous models audio tagging models. However, in the context of visual benchmarks such as ImageNet, open-ended text-visual models such as CLIP have previously outperformed domain expert models such as various types of CNNs. A similar under-performance can be seen in the few-shot audio classification task (Table 2), especially for the 12-way case. The authors try to attribute this poor performance to the length of the input, but they did not explain why the model couldn't be configured and trained with a sufficiently long input context window in order to accommodate this task and thereby address this limitation.
W2. The section on integration of BLIP-2 and APT-LLM (Section 4.4) is not easy to follow for readers not familiar with the audio-visual tasks. Not enough background information is provided about either the BLIP-2 model architecture or the audio-visual learning task that the author performed evaluation on. This unfortunately makes this part of the claimed contribution less convincing.
W3. The natural-language audio reasoning (NLAR) dataset is constructed from a subset of the Clotho-AQA dataset by utilizing OpenAI's ChatGPT-turbo API. The NLAR dataset suffers from two issues: 1) the authors did not explain the criteria by which the subset was selected from Clotho-AQA, and 2) the authors did not describe how the quality of the examples generated by ChatGPT-turbo was controlled. Presumably, some sort of manual inspection was required to ensure that the LLM-generated test examples are correct.

**Questions:**

Q1. The diagram in Figure 1 seems to miss a part between the input text and the audio aligner for tokenizing the input text and looking up the embeddings for the input text for the aligner's use. This figure needs clarification.
Q2. How does the frozen LLM receive the audio-text juxtaposed embeddings? Its built-in embedding lookup layer should have been removed so that the mixed audio-text embeddings can be passed directly as the input to the LLM. This begs the question of why the authors decided to freeze the LLM. It seems that by freezing the LLM, the burden and opportunities for learning are limited to the audio encoder and the audio aligner. Is it possible allowing the weights of the LLM to change and adapt during training would lead to better learning outcomes by the entire APT-LLM model? The authors didn't describe any hyperparameter tuning processes.
Q3. Equation (1) and Equation (4) seem to have inconsistency with Figure 1. M_{\theta} should take an additional input (the text) according to the diagram in Figure 1.
Q4. How many examples does the NLAR dataset contain?
Q5. Figure 3 in the appendix lacks legends. In addition, the y-axis on the right-hand side is unlabeled. As a result, it is unclear what are plotted by the blue and orange curves, which makes this figure hard to read.

---

> ### Author Response · Authors · 2023-11-23
> **Response to Reviewer cBEu**
>
> Thank you for your valuable feedback! It helps us to improve the quality of the paper. We address your comments one by one as follows:
>
> [Experiment on audio tagging and captioning]
>
> For the audio classification, we found that there is a performance drop due to the difficulty for the open-set problem and text matching. For example, when CLAP, a open-set classifier pretrained with contrastive loss, is used for evaluation, it calculates the similarity with 527 classes whereas APT predicts an appropriate token out of 32k tokens. It is harder for generative models to carry out multi-label classification where the threshold is essential to predict the presence of a sound event.
>
> Another issue is how to match the predicted answers with the ground-truth labels. Suppose APT outputs “rat” and we get the class of “mouse” in the ground truth. In that case, a proper text matching approach is required for a fair comparison. During evaluation on audio classification tasks, we used the APT’s checkpoint from audio-text-alignment pretraining as the text encoder. However, we did find that using different text encoders to evaluate the same checkpoint impacts the classification values a lot. We are planning to apply some discriminative text encoders, such as CLIP text encoder and ChatGPT Ada model, for the evaluation, and we will add the discussion on text matching to the Appendix.
>
> Regarding the audio captioning task, training on multiple datasets does not improve the performance on a specific dataset because the model cannot fit the dataset-specific vocabulary. In addition to the general-purpose audio [1], we found the same phenomena in other domains [2,3]. Following [3,4], we have conducted additional experiments to finetune APT on both Clotho and AudioCaps datasets for two epochs only. We compared the finetuned APT with expert models trained on the same datasets. As shown in Table 4, the APT yielded the best scores on both tasks.
>
> [Background of the audio-visual learning task]
>
> Thanks for your feedback. To improve the readability, we have added the introduction on both BLIP-2 structure and the audio-visual task to Section 4.4.
>
> [Experiment on few-shot learning]
>
> Due to the limited computational resource, we didn’t extend the contextual windows to meet the few-shot settings. We are now re-training the models with a longer max length of sequence and will update the paper accordingly.
>
> [missing tokenizer]
>
> Thanks for pointing that out. e have now updatedFigure 1.
>
> [Answer to Q2]
>
> Actually, the LLM’s embedding layer was not removed in APT-LLM. Instead, we truncated the input sequence as per positions of audio indicators and concatenated them again once acoustic/text tokens have been encoded (separately).
>
> Regarding the reason why we froze the LLM, the motivation of this work is to extend the existing LLMs/VLMs to the audio domain and prevent it from losing its original abilities as an LLM/VLM. To this end, instead of adjusting the LLM, we learn an audio aligner to generate soft prompts, conditioned on audio-text embeddings, for the language model. While finetuning LLMs with PEFT will boost the overall performance, we argue that it may jeopardise the LM performance on original tasks, even causing catastrophic forgetting. By tuning audio aligner only, we build up an audio language model with fewer parameters and fewer training samples while yielding a comparable performance on each understanding task compared to other audio foundational models.
>
> [the NLAR task]
>
> Thank you for pointing that out. We will add a more detailed introduction to the Appendix A.3.
>
> [Missing legend in Fig. 3]
>
> Thank you for your comment. We will update the revised one in the latest version of the manuscript.
>
>
> [1] PREFIX TUNING FOR AUTOMATED AUDIO CAPTIONING
>
> [2] LLARK : A MULTIMODAL FOUNDATION MODEL FOR MUSIC
>
> [3] SIMVLM: SIMPLE VISUAL LANGUAGE MODEL PRETRAINING WITH WEAK SUPERVISION
>
> [4] BLIP-2: Bootstrapping Language-Image Pre-training with Frozen Image Encoders and Large Language Models

---

### Official Review · Reviewer_TGqa · 2023-11-05

**Soundness:** 3 good
**Presentation:** 3 good
**Contribution:** 3 good
**Rating:** 8
**Confidence:** 3

**Summary:**

The paper introduces Acoustic Prompt Turning (APT), an acoustic adapter that extends large language models (LLMs) and visual language models (VLMs) to the audio domain. APT uses an instruction-aware aligner to acquire acoustic embeddings from audio feature maps, allowing it to handle diverse audio-related tasks in a sequence-to-sequence manner. The paper demonstrates the effectiveness of APT-LLMs through various tasks and introduces a novel audio reasoning task. It also shows that APT can extend frozen VLMs to the audio domain, yielding promising results in the audio-visual understanding task.

**Strengths:**

- The concept of APT is innovative and presents a new direction for extending LLMs and VLMs to the audio domain without compromising their domain-specific capacity. This also provides evidence that encoding sound clips as word tokens is an efficient approach to adapt LLM/VLM to the audio domain.
- Introducing the natural language audio reasoning task is a creative way to evaluate model's ability to understand, compare, and summarise two audio clips.
- The paper does a good job comparing its work with existing models, providing a clear context for the novelty and utility of APT.
- There are significant performance improvements across audio-visual baselines, highlighting the effectiveness of APT in the audio-visual domain. The performance on most of the open-ended tasks was good.

**Weaknesses:**

- Performance on Certain Tasks: Despite the novelty of the idea, the performance of APT-LLMs on the close-ended datasets - ESC50 (few shot classification) and AudioSet (captioning) tasks is not competitive compared to state-of-the-art, task-specific models. This indicates a need for improvement in these areas.
- Handling of Errors: It is unclear how the model handles potential discrepancies or errors in the labeled examples used for in-context learning. This raises questions about the robustness of the model in real-world scenarios where such issues are common.

**Questions:**

- How can APT's performance on tasks like ESC50 and AudioSet be improved to be more competitive with task-specific models?
- How does the instruction-aware aligner in APT handle different audio signals, especially those with complex characteristics?
- How can the model be improved to handle discrepancies or errors in the labeled examples used for in-context learning?
- How does APT perform for more specific types of audio (e.g., music, notes...)

---

> ### Author Response · Authors · 2023-11-23
> **Response to Reviewer TGqa**
>
> Thank you for your enlightening feedback! We address your comments one by one as follows:
>
> [Performance on certain audio tasks]
>
> We notice that compared to the common classifiers, there is a performance gap due to the difficulty for the open-set problem and text matching. For example, when CLAP, a open-set classifier pretrained with contrastive loss, is used for evaluation, it calculates the similarity with 527 classes whereas APT predicts an appropriate token out of 32k tokens. Another issue is how to match the predicted answers with the ground-truth labels. Suppose APT outputs “rat” and we get the class of “mouse” in the ground truth. In that case, a proper text matching approach is required for a fair comparison. During evaluation on audio classification tasks, we used APT’s checkpoint from audio-text-alignment pretraining as the text encoder. However, we did find that using different text encoders to evaluate the same checkpoint impacts the classification values a lot. We are planning to apply some discriminative text encoders, such as CLIP text encoder and ChatGPT Ada model, for the evaluation, and we will add the discussion on text matching to the Appendix.
>
> Regarding the audio captioning task, training on multiple datasets does not improve the performance on a specific dataset because the model cannot fit the dataset-specific vocabulary. In addition to the general-purpose audio [1], we found the same phenomena in other domains [2,3]. Following [3,4], we have conducted additional experiments to finetune APT on both Clotho and AudioCaps datasets for two epochs only. We compared the finetuned APT with expert models trained on the same datasets. As shown in Table 4, the APT yielded the best scores on both tasks.
>
> [Error handling]
> We can roughly divide the errors into two types:
> a) label error (noisy label): It is actually really interesting to notice in our preliminary study that if we replaced the name of sound event with an abstract but distinguishable one, e.g., “dog” -> “class A”, APT will still figure out the most similar one in terms of acoustic characters and return with “class A”.
> b) audio discrepancy: this is really a good question. To be honest, we didn’t set up experiments for that so far. Yet, it is very interesting for future work. We have added our discussion on error handling for the future work at the end of the manuscript.
>
> [Handling different audio signals]
> The instruction-aware aligner processes audio signals by extracting the relevant audio features according to input text. Therefore, in addition to the audio characteristics, it relies on prompts as well.
> Regarding more specific types of audio, we are planning to extend APT to speech and music as well. Since we use AudioMAE, a generative model trained with reconstruction loss, as the audio encoder, APT has the potential to generalise to recognise speech and music.
>
> [1] PREFIX TUNING FOR AUTOMATED AUDIO CAPTIONING
>
> [2] LLARK : A MULTIMODAL FOUNDATION MODEL FOR MUSIC
>
> [3] SIMVLM: SIMPLE VISUAL LANGUAGE MODEL PRETRAINING WITH WEAK SUPERVISION
>
> [4] BLIP-2: Bootstrapping Language-Image Pre-training with Frozen Image Encoders and Large Language Models

---

### Official Review · Reviewer_w4rk · 2023-11-05

**Soundness:** 3 good
**Presentation:** 3 good
**Contribution:** 3 good
**Rating:** 6
**Confidence:** 3

**Summary:**

The paper introduces Acoustic Prompt Turning (APT), an acoustic adapter that extends large language models (LLMs) and visual language models (VLMs) to the audio domain. Existing models have limited applicability to audio tasks.

APT uses a multi-task learning framework and an instruction-aware aligner to generate fixed acoustic embeddings from audio feature maps. Various audio-related tasks are formulated in a sequence-to-sequence manner, allowing APT to be trained without constraints on input sequences. Experimental results show that LLMs coupled with APT achieve competitive performance compared to expert models across different tasks. APT is also evaluated on a novel audio reasoning task and shown to extend frozen VLMs to the audio domain, even without fine-tuning on audio-visual datasets.

**Strengths:**

1. This paper pioneers the exploration of audio-text language modeling, particularly in addressing the format constraint that previous work faced. In the past, there have been a few attempts at audio-text foundation modeling; however, they were limited to the input format [audio, Q, A], excluding support for other practical tasks. As a result, these models were unable to exhibit the same level of intelligence as popular language models like GPT-4. This paper overcomes this limitation by considering audio as a prompt in language modeling, enabling it to perform various tasks.

2. Additionally, the paper introduces a novel task called audio reasoning and provides a dataset that will prove invaluable for future research. This direction is highly significant as existing datasets often prove too simplistic for large models, failing to capture the complexity and intelligence required in real-world audio modeling scenarios. By introducing a more challenging audio reasoning task and accompanying dataset, the paper paves the way for the development of smarter and more sophisticated audio models that better align with real-world demands.

**Weaknesses:**

1. When examining the experimental results, it is apparent that the proposed model does not perform as strongly as the specific model on certain tasks. For instance, the baseline for AudioSet classification is around 47, whereas this paper only achieves 14.7. If a foundation model lags behind a specific model, its technical significance is limited. Furthermore, the model does not demonstrate sufficient strength in the audio caption task, which should ideally be robust considering the capabilities of the LLM as a decoder.

The authors should provide evidence to support the advantages of foundation models. If a foundation model is merely capable of performing multiple tasks, it may not be sufficient. It is important for the authors to demonstrate the unique strengths and benefits of the foundation model compared to other approaches. This could include showcasing improved performance, increased efficiency, or enhanced generalization across tasks. Providing such evidence will help establish the significance and value of the foundation model in the audio domain.

2.Upon examining the approach outlined in the paper, it becomes evident that it is similar to  existing speech-language models. Initially, when reading the title of the paper, I anticipated that the method would differ significantly from previous models such as AudioPaLM and SpeechLM. However, upon closer inspection, it appears that the method aligns closely with these established models, but changing the input from speech to audio. Given the current era of large language models (LLMs), my expectations for groundbreaking innovations were not high.

**Questions:**

1. How to gurantee the audio reasoning dataset qualilty? It is created by ChatGPT and may need some human participants for quality check.

---

> ### Author Response · Authors · 2023-11-23
> **Response to Reviewer w4rk**
>
> Thank you for your valuable feedback! It is very helpful to improve the quality of the paper. We address your comments one by one as follows:
>
> [Experiments on common audio tasks]
>
> We notice that compared to the common classifiers, there is a performance drop due to the difficulty for the open-set issue and text matching. For example, when CLAP, a open-set classifier pretrained with contrastive loss, is used for evaluation, it calculates the similarity with 527 classes whereas APT predicts an appropriate token out of 32k tokens. Another issue is how to match the predicted answers with the ground-truth labels. Suppose APT outputs “rat” and we get the class of “mouse” in the ground truth. In that case, a proper text matching approach is required for a fair comparison. During evaluation on audio classification tasks, we used the APT’s checkpoint from audio-text-alignment pretraining as the text encoder. However, we did find that using different text encoders to evaluate the same checkpoint impacts the classification values a lot. We are planning to apply some discriminative text encoders, such as CLIP text encoder and ChatGPT Ada model, for the evaluation, and we will add the discussion on text matching to the Appendix.
>
> Regarding the audio captioning task, training on multiple datasets does not improve the performance on a specific dataset because the model cannot fit the dataset-specific vocabulary. In addition to the general-purpose audio [1], we found the same phenomena in other domains [2,3]. Following [3,4], we have conducted additional experiments to finetune APT on both Clotho and AudioCaps datasets for two epochs only. We compared the finetuned APT with expert models trained on the same datasets. As shown in Table 4, the APT yielded the best scores on both tasks.
>
>
> [Advantages of foundational models]
>
> Audio foundational models highlight in two-fold:
>
> Fast adaptation: Audio foundational models fit a new dataset-specific scenario promptly by leveraging task homogeneity. For instance, rather than learning from scratch, a foundational model has already done a very good job on feature extraction and thus learns to understand the abstract concepts only. This can be supported by our experiments: when learning from the natural language audio reasoning task, it only requires roughly 1-hour audio data to train APT as it has already learned to capture acoustic patterns from a series of understanding tasks.
>
> Improved performance: Task diversity is important for audio foundational models to learn better audio feature representation compared to other models. As shown in Table 4, the APT-LLM yields the best results on both datasets, outperforming the state-of-the-art models.
>
>
> [Comparison between AudioPALM and SpeechLM]
>
> AudioPALM and SpeechLM are great works bridging speech and text modalities. They expanded textual vocabulary to the speech-text one as both modalities share a basic unit - phonemes. They then mix the multi-modal tokens together and thus succeed in cross-modality transition. However, general-purpose audio language models feature the challenge that the basic units, namely sound events, are unstructured, sparsely distributed and often overlapping, sharing very little common place with phoneme-level text units. Moreover, the tasks solved in this work need more discriminative information compared to speech-text tasks, calling for a robust audio encoder and text-audio alignment.
>
> [Regarding data curation]
>
> Thanks for pointing that out. We actually manually cleaned the data after curating them automatically. We will add this content to the Appendix.
>
> We appreciate your valuable feedback. We will update the content we have discussed above to the manuscript.
>
> [1] PREFIX TUNING FOR AUTOMATED AUDIO CAPTIONING
>
> [2] LLARK : A MULTIMODAL FOUNDATION MODEL FOR MUSIC
>
> [3] SIMVLM: SIMPLE VISUAL LANGUAGE MODEL PRETRAINING WITH WEAK SUPERVISION
>
> [4] BLIP-2: Bootstrapping Language-Image Pre-training with Frozen Image Encoders and Large Language Models

---

### Official Review · Reviewer_u172 · 2023-11-07

**Soundness:** 2 fair
**Presentation:** 2 fair
**Contribution:** 2 fair
**Rating:** 1
**Confidence:** 2

**Summary:**

The authors study the work of empowering large language models with audition capabilities. However, the idea and the presentation of this paper is very similar to BLIP-2: Bootstrapping Language-Image Pre-training with Frozen Image Encoders and Large Language Models (https://arxiv.org/pdf/2301.12597.pdf), we can see that the Figure 2 in this paper is very similar to Figure 2 in BLIP-2, the losses and masking strategies and so on are very similar to BLIP-2. To empower large language models with audition capabilities, the authors should propose some new idea. In the experiments, the authors don't compare their work with some well-known approaches such as SpeechGPT and so on.

**Strengths:**

1. The problem is interesting.

**Weaknesses:**

1. The Figure 2 in this paper is very similar to Figure 2 in BLIP-2: Bootstrapping Language-Image Pre-training with Frozen Image Encoders and Large Language Models (https://arxiv.org/pdf/2301.12597.pdf), just replace the word image to audio. Thus, the novelty of this paper is very limited compared with BLIP-2. The Figure 1 in this paper is similar to Figure 3 in BLIP-2. The authors use the exactly three loss in BLIP-2, matching loss, contrastive loss and gounded text generation loss, with the exactly three mask strategies, and the authors use the learnable query.

2. Model Capability: Unlike methods such as SpeechGPT, the approach presented in this article limits the use of speech modality to input only, preventing the synthesis of speech output. This results in a model lacking genuine speech interaction capability.

3. Experimental Comparisons and Results: The performance of the method falls below the expected standard, and there is a notable absence of performance comparison with established works such as SpeechGPT.

**Questions:**

1. The audio modality is different from image modality. Why the authors use the exactly thress losses and three mask strategies from Figure 3 in BLIP-2 ?
2. Apart from the novelty, in the experiments, the authors don't compare their model with some well-known works, such as SpeechGPT and so on.
3. For the audio modality, speech interaction is important, why not the authors focus on this part, which is different from image modality.

---

> ### Author Response · Authors · 2023-11-23
> **Response to Reviewer u172**
>
> Thank you for your valuable feedback! We address your comments one by one as follows:
>
> [Motivation]
>
> To clarify, proposing a new network architecture for LLMs is not the aim of this work. Instead, as stated in the introduction, our goal is to extend LLMs and VLMs to the audio domain and further explore a more general generative audio-language framework that ingests as inputs multiple audio clips in the inference while maintaining their ability as LLMs or VLMs. Our experiments verified that the proposed generative framework is a promising direction and we hope it may be of interest for future multi-modal LM research. We will update our manuscript to better present our motivation and methodology.
>
> [APT v.s. BLIP-2]
>
> Motivated by BLIP-2’s idea on Qformer, this work applies a BERT-ish transformer to bridge the modality gap. After the preliminary study, we found the combination of contrastive loss, grounded loss, and language generative loss, as depicted in BLIP-2, performed the best to pretrain the transformer. We also adjusted the number of transformer blocks and dimension of hidden states for a better performance of audio alignment. While BLIP-2 pretrained this transformer with the captioning task only, this work combines captioning and tagging tasks to mitigate the lack of high-quality data in the audio domain.
>
> The distinction of this work is to learn a more versatile framework that is capable of learning from more diverse tasks and can easily adapt LLMs/VLMs to the audio modality. As shown in Fig. 1, this work highlights a trainable audio identification token added to each group of acoustic tokens from the audio aligner and the juxtaposition between grouped acoustic tokens and text embeddings. This enables the proposed framework to address multiple diverse tasks, such as few-shot learning and complex audio understanding, in both training and inference stages compared to BLIP-2. Moreover, to evaluate more rigorously how the proposed model benefits from LLMs beyond common audio tasks, we proposed a new task, namely natural language audio reasoning, that tests the logic capacity of models with a series of comparison and summarization tasks.
>
> [APT v.s. SpeechGPT]
>
> SpeechGPT is a great work that integrates speech recognition and generation within one architecture. However, the focus of SpeechGPT is speech recognition/generation which is quite different from this work where an adapter is studied to extend LLMs/VLMs to general-purpose audio understanding tasks.
>
> We acknowledge that speech processing is essential to AI development, but perceiving the general audio is also challenging. For example, different from speech with well-known phoneme structure, basic units of environmental audio, namely sound events, are distributed more sparsely without a clear structure, and they are more prone to overlap with each other.
>
> However, inspired by the idea of SpeechGPT, this work can be extended to explore foundational models for joint understanding and generation of general audio in the future.

---

### Meta-Review · Area_Chair_f1cm · 2023-12-11

**Metareview:**

This paper was reviewed by four experts in the field and received a mixed score. The main concerns are the limited novelty, unconvincing experiments, and lack of clarity. The authors did a good job of rebuttal and addressed many of the concerns. However, the reviewers (including all positive ones) still feel that more work is needed to get it to the best version. The main concerns include:

1. The motivation and contribution of this paper are not well illustrated.
2. The results are not good enough.
3. The clarity of the presentation needs to be improved.


AC also agrees that this work can be much stronger with additional experiments. While this paper clearly has merit, the decision is not to recommend acceptance. The authors are encouraged to consider the reviewers' comments when revising the paper for submission elsewhere.

**Justification For Why Not Higher Score:**

1. The motivation and contribution of this paper are not well illustrated.
2. The results are not good enough.
3. The clarity of the presentation needs to be improved.

**Justification For Why Not Lower Score:**

na

---

### Decision · Program_Chairs · 2024-01-16

Reject